# Defect-Induced Gas-Sensing Properties of a Flexible SnS Sensor under UV Illumination at Room Temperature

**DOI:** 10.3390/s20195701

**Published:** 2020-10-07

**Authors:** Nguyen Manh Hung, Chuong V. Nguyen, Vinaya Kumar Arepalli, Jeha Kim, Nguyen Duc Chinh, Tien Dai Nguyen, Dong-Bum Seo, Eui-Tae Kim, Chunjoong Kim, Dojin Kim

**Affiliations:** 1Department of Materials Science and Engineering, Chungnam National University, Daejeon 34134, Korea; hungnm@lqdtu.edu.vn (N.M.H.); chinhnd@cnu.ac.kr (N.D.C.); Sdb987@cnu.ac.kr (D.-B.S.); etkim@cnu.ac.kr (E.-T.K.); 2Department of Materials Science and Engineering, Le Quy Don Technical University, Hanoi 100000, Vietnam; Chuong.vnguyen@lqdtu.edu.vn; 3Department of Energy Convergence Engineering, Cheongju University, Cheongju 28503, Korea; vinaykumararepalli@gmail.com (V.K.A.); jeha@cju.ac.kr (J.K.); 4Institute of Theoretical and Applied Research, Duy Tan University, Hanoi 100000, Vietnam; nguyentiendai@duytan.edu.vn; 5Faculty of Natural Sciences, Duy Tan University, Da Nang 550000, Vietnam

**Keywords:** SnS thin film, vacancy, flexible sensor, UV illumination, density functional theory

## Abstract

Tin sulfide (SnS) is known for its effective gas-detecting ability at low temperatures. However, the development of a portable and flexible SnS sensor is hindered by its high resistance, low response, and long recovery time. Like other chalcogenides, the electronic and gas-sensing properties of SnS strongly depend on its surface defects. Therefore, understanding the effects of its surface defects on its electronic and gas-sensing properties is a key factor in developing low-temperature SnS gas sensors. Herein, using thin SnS films annealed at different temperatures, we demonstrate that SnS exhibits n-type semiconducting behavior upon the appearance of S vacancies. Furthermore, the presence of S vacancies imparts the n-type SnS sensor with better sensing performance under UV illumination at room temperature (25 °C) than that of a p-type SnS sensor. These results are thoroughly investigated using various experimental analysis techniques and theoretical calculations using density functional theory. In addition, n-type SnS deposited on a polyimide substrate can be used to fabricate high-stability flexible sensors, which can be further developed for real applications.

## 1. Introduction

The era of the Internet of Things has resulted in the need to integrate various functions into portable and flexible devices. In terms of such devices, the development of sensors with hazardous gas-detecting ability to ensure safe and healthy living and work spaces is an interesting and vital undertaking. Current gas sensors are mainly based on metal oxide semiconductors (MOSs) because of their high response and good stability. However, a significant limitation of MOS sensors is the high operating temperatures (100–600 °C) they require [1,2,3]. Therefore, it is not suitable to integrate current MOS sensors into battery-powered devices because of the high power consumption such temperatures require. Furthermore, the development of wearable, stretchable, and flexible devices, which is becoming increasingly important for modern technology, necessitates the use of flexible substrates (e.g., polyimide (PI) or polyethylene terephthalate (PET)). However, the decomposition temperatures of such substrates are low. In addition, a significant difference in the coefficient of thermal expansion between MOS-sensing layers and flexible substrates is a limitation in the application of MOS-sensing layers.

Several strategies have been proposed to solve the issues, including the modification of MOS nanostructures, ultraviolet (UV) and/or visible light illumination, and doping to reduce their operating temperatures. However, the development of low-temperature flexible MOS sensors still encounters an obstacle, which is its strong deformation under heat treatment. Therefore, development of new gas-sensing materials operating with low-temperature heat treatment is required. Lately, transition metal chalcogenides (TMCs) have attracted attention for sensing applications owing to their reversible sensing ability at low temperatures, even at room temperature (RT) [4,5,6,7]. However, the sensitivity and response/recovery times of chalcogenide sensors need to be improved if they are to be applied in practical sensors. Experimental and theoretical studies have revealed that the gas-sensing behavior of TMC-based sensors depends strongly on the cation and anion vacancies of TMCs [8,9,10]. Therefore, controlling such vacancies is considered a key strategy for enhancing the low-temperature detection properties of TMC sensors [9]. In addition to such defect control, the use of ultraviolet (UV) illumination is an efficient way of enhancing the response while shortening the recovery time of TMC sensors [11]. Accordingly, a combined strategy involving both UV illumination and vacancy control is expected to significantly enhance the performance of TMC sensors.

Tin sulfide (SnS) is a TMC material with natural p-type properties that originate from an excess of S atoms. However, the presence of S vacancies has been reported to impart n-type properties to SnS by some authors [12,13]. From the viewpoint of gas-sensing applications, p-type SnS sensors show the ability to detect toxic gases and volatile organic compounds (VOCs) at RT [6,14]. The disadvantage of p-type SnS sensors is their high base resistance, which hinders the ability to integrate them into portable and flexible devices. By contrast, n-type SnS is a good candidate for an RT sensing layer because of its lower resistance [15]. Moreover, the extremely low deformation of SnS thin-film on PI substrate under heat treatment (as discussed later) is a great advantage in developing a flexible sensor. However, the gas-sensing properties of n-type flexible SnS sensors have not been investigated to date, although the fundamental chemical and physical properties of n-type SnS have been widely investigated [12,13].

In this study, we have demonstrated that SnS thin films synthesized by sputtering exhibit high adhesion to PI substrates. Furthermore, SnS thin film-based flexible sensors can withstand heat up to 300 °C without deformation and work stably under various bending conditions and UV illumination at RT. Moreover, a comparison of the sensing performances of p-type and n-type SnS sensors revealed that the gas-sensing performance of SnS sensors strongly depends on the type of vacancies in the SnS sensing films. These insights could provide an efficient strategy for constructing n-type SnS-based nanostructures for high-performance RT gas sensors.

## 2. Experimental Section

### 2.1. Sensor Fabrication

SnS thin films with different thicknesses (30, 50, 80, and 100 nm) were deposited onto PI substrates using radio frequency (RF) sputtering from an SnS target. The substrate target distance was fixed at 6.8 cm, while the sputtering chamber pressure was maintained at 30 mTorr with a 30 sccm air flow and a vacuum pump. The substrate temperature was maintained at 200 °C during the deposition process. Various thicknesses of SnS thin films were obtained by changing the deposition time. The as-deposited samples with different thicknesses were labeled SnS-30, SnS-50, SnS-80, and SnS-100, where the number in the label represents the thickness of the SnS layer in nm. After the sputtering process, some thin films were heated at different temperatures (250, 300, and 400 °C) for 1 h in air. These samples are denoted by SnS-xxx-Hxxx, where Hxxx indicates the heating temperature (e.g., SnS-30-H300 represents an SnS thin film with a thickness of 30 nm heated at 300 °C). Therefore, the effects of surface defects on the morphology, structure, and gas-sensing properties of SnS thin films were investigated. The SnS sensors were fabricated by depositing a platinum comb electrode on the surface of the corresponding SnS thin film using a DC sputtering technique and a shadow mask (Figure 1 and Appendix A). Similar thin films were also deposited onto glass substrates for TEM analysis, Hall effect measurements, and absorbance measurements.

### 2.2. Material Characterization and Gas-Sensing Measurement

The surface morphologies of the SnS thin films were investigated by scanning and transmission electron microscopies (SEM, TEM) and atomic force microscopy (AFM), using JSM 700F (SEM), JEM-ARM 200F (TEM), and XE7-Park (AFM) systems. Crystalline structures were evaluated by analyzing the data obtained from selected area electron diffraction (SAED) using TEM as well as X-ray diffraction using a D8 Discover diffractometer with parallel and monochromatic Cu Kα radiation (λ = 0.15418 nm) formed by Göbel mirrors and a Ge (022) channel-cut monochromator. In addition, room-temperature Raman spectroscopy with a 532 nm laser source (SR 303i) was conducted to further investigate the presence of phases in SnS thin films. The chemical states of the elements were determined by X-ray photoelectron spectroscopy (XPS) using a VG Multilab 2000 system. XPS peaks were calibrated using the binding energy of C 1s (284.8 eV), while the Sn/S atomic ratio was calculated from the areas of the respective characteristic peaks (Sn 3d and S 2p) using Thermo Avantage software. The calculation was adjusted with relative sensitivity factors (RSF) of 37.257 and 1.881 for Sn 3d and S 2p, respectively. The major carrier concentration and the carrier type (electron or hole) were determined by Hall effect measurement on an Ecopia HMS 5300 system at RT. The absorbance spectra of the thin films were recorded to compare the bandgaps of SnS-80 thin films before and after annealing at 300 °C (SnS-80-H300) using an UV-Vis spectrophotometer (UV-2600, Shimadzu, Co., Ltd.).

NO_2_-sensing performances in the dark and under UV illumination were investigated by recording the real-time resistance of the sensors using a gas-sensing measurement setup, which is similar to that used in our previous studies [16,17,18]. In this study, a radiation of 254 nm wavelength with a constant intensity of 12.7 mW was used to illuminate the surface of the sample during the gas-sensing measurement process. The response (S) of the sensor was defined as S = (R_g_ − R_a_/R_a_) × 100% or S = (R_a_ − R_g_/R_a_) × 100% when the resistance of the sensor increased or decreased in the presence of the target gas, respectively. Here, R_o_ and R_g_ are the corresponding resistance values of the sensor in dry air and that in a mixture of the target gas and dry air, respectively. The recovery time of the sensor was defined as the time taken to reach ~90% decay from the response level, whereas the response time was defined as the time taken to reach a ~90% maximum response value.

### 2.3. Density Functional Theory Calculations

Calculations were performed using density functional theory (DFT) based on first-principles calculations. To describe the exchange and correlation energies, we used the generalized gradient approximation (GGA) and Perdew–Burke–Ernzerhof (PBE) approximations within the projector augmented wave (PAW) pseudopotentials, which are implemented in the Quantum Espresso simulation package. For all the structural optimizations and electronic properties calculations, the cutoff energy was set to 510 eV, and an (8 × 8 × 1) k-point mesh was used to sample the Brillouin zone (BZ). All of the structural geometry was fully relaxed until the total energies converged to below 10^−6^ eV and the forces acting on the atoms were less than 10^−3^ eV/Å. A large vacuum thickness of 30 Å along the z-direction of the system was applied to avoid interactions between neighboring slabs. Furthermore, to examine the effects of gas adsorption, we used a (4 × 4 × 1) supercell of pure SnS and defective SnS monolayers.

## 3. Results and Discussion

### 3.1. Morphological and Structural Characteristics

Figure 2a–d shows the surface SEM images of the as-deposited SnS thin films with their corresponding thicknesses shown in the inset. The thicknesses of the thin films estimated from the cross-sectional SEM images were approximately 30, 50, 80, and 100 nm. Furthermore, the composition and homogeneous distribution of Sn and S elements in a SnS thin film deposited on a glass substrate were confirmed by elemental mapping using TEM (as will be shown later in Figure 6e). Since the temperature of the holder remained at 200 °C during the deposition of SnS thin films, the particle sizes and porosities of the thin films are significantly dependent on deposition time. The SnS-30 thin film was composed of fine SnS particles (~10 nm) with low porosity, whereas the SnS-100 thin-film showed coarser particles (~60 nm), thereby leading to greater pore sizes. Generally, the deposition time strongly affects the morphologies and porosities of the as-deposited thin films. The longer the deposition time, the larger the particle and the higher the porosity. The SEM results are in good agreement with the results observed from the AFM images (Figure 3a–d). The surface roughness of the SnS thin films was proportional to their thicknesses. The roughness values were 1.1, 1.8, 2.1, and 3.2 nm for SnS-30, SnS-50, SnS-80, and SnS-100 thin films, respectively. In terms of crystal structure, Figure 2e indicates that all as-deposited SnS thin films exhibited orthorhombic lattice structures (JCPDS, # 00-053-0526). The more intense peaks are ascribed to the (012), (110), (004), and (113) crystalline planes. Among them, the (012), (110), and (400) planes were also confirmed by SAED (Figure 6b), where the characteristic diffraction circles indicate the polycrystalline structure of the Sn-S80-H300 thin film. The high-resolution TEM image (Figure 6c) also revealed the polycrystalline structure of the SnS thin film, which is appropriate for the SEAD results shown. Therefore, all the prepared SnS thin films were polycrystalline. In the X-ray diffraction (XRD) patterns in Figure 2e, two small peaks characteristic of hexa-sulfur (S6, JCPDS, 01-086-2289) can also be observed, indicating the decomposition of SnS thin films during deposition at 200 °C. However, other Sn_x_S_y_ phases (e.g., SnS_2_ and Sn_2_S_3_) were not detected under the resolution of the XRD equipment. The Raman results further confirm the presence of possible phases in the as-deposited SnS thin films (Figure 2f). In general, all of the SnS thin films were confirmed by peaks indexed at 98 and 229 cm^−1^, which are assigned to Ag vibration [19,20,21,22]. Typical Raman peaks of S6 were not detected, suggesting its negligible presence. It should be noted that the strongest peak for SnS_2_ at 317 cm^−1^ was observed with a weak intensity in the SnS-50 thin film [21,23,24]. However, this peak was not observed in the other samples. These observations indicate that the appearance of the SnS_2_ and S6 phases was negligible, random, and fluctuating. In general, S6 and SnS_2_ phases do not always exist, and they do not significantly affect the properties of SnS thin films, as shown in our previous report [25] and this study (as shown later). From the XRD and Raman spectra, we conclude that the as-deposited SnS thin films are composed of an almost orthorhombic SnS phase and an insignificant, random amount of S6 and SnS_2_ phases. The negligible presence of these phases does not significantly affect the electrical and gas-sensing properties of SnS thin films in general. Hall effect measurements, absorbance measurements, and gas-sensing measurements further support this assessment, as shown later.

Figure 4a–c shows the morphology of SnS-80 thin-film annealed at different temperatures (250–400 °C). The particle-size growth was more rapid when the annealing temperature was increased from 250 to 400 °C. The crystalline sizes of SnS-80, SnS-80-H250, and SnS-80-H300 estimated by Scherrer equation were 10.2, 17.5, and 18.2 nm, respectively (Appendix A). These values are smaller than the corresponding particle sizes observed by SEM and AFM. Therefore, SnS particles are polycrystals rather than single crystals. In addition to particle-size growth, the surface roughness of the SnS-80 thin film was proportional to the annealing temperature (Figure 3c,e,f). There were no new phases formed in all of the SnS-80 thin films annealed below 300 °C, as confirmed by XRD and Raman spectra (Figure 4d,e). The intensities of the Raman peaks of SnS (98 and 229 cm^−1^) were weak, while other strong peaks at approximately 161 and 192 cm^−1^ were not observed [26]. The results indicate the low crystallinity of the SnS thin films, although these thin films were annealed at high temperatures. This assessment is consistent with the high-resolution TEM images (Figure 6c,d).

We also compared the bandgaps of SnS-80 and SnS-80-H300 using Tauc plots inferred from absorbance curves (Figure 4f). The results indicate that the former had a bandgap of 1.75 eV, whereas the bandgap of the latter was estimated to be 1.5 eV. These results are consistent with previous literature reports [25,27,28]. The decrease in the bandgap value of SnS-80-H300 originates from the increase in particle size, as confirmed by SEM (Figure 2c and Figure 4b) and AFM images (Figure 3c,f). It should be noted that the bandgap of SnS_2_ is approximately 2.18–2.75 eV [27,29,30,31]. Absorbance measurement is a bulk analysis technique. Therefore, these results allow us to safely conclude that the presence of SnS_2_ and S6 phases, as discussed above, is negligible within all of the SnS thin films annealed below 300 °C. By contrast, the SnS-80-H400 thin film exhibited many different Sn_x_S_y_ phases, including SnS, Sn_2_S_3_, and SnS_2_ (Figure 4d). The Raman spectrum also showed the strongest Raman peak (317 cm^−1^) of the SnS_2_ phase in the SnS-80 thin film annealed at 400 °C (Figure 4e). The formation of SnS_2_ at high temperatures in the air was also reported elsewhere [15]. It has been noted that a PI substrate begins to decompose significantly at temperatures over 350 °C [32]. This study focused on evaluating the gas-sensing properties of SnS thin films on PI substrates. Therefore, the samples annealed below 300 °C were chosen for further investigation.

In order to determine the contaminant levels and valence states of the elements in SnS thin films annealed below 300 °C, XPS profiles were also obtained, and the results are shown in Figure 5. All the survey spectra of the SnS thin films, including SnS-80, SnS-80-H250, and SnS-80-H300, displayed characteristic binding energy levels for Sn and S elements, whereas no peaks for contaminants are observed (Figure 5a). The high-resolution Sn 3d spectrum exhibited two characteristic peaks at 493.63 and 485.13 eV, which are ascribed to the Sn 3d_3/2_ and Sn 3d_5/2_ energy states of Sn^2+^ in SnS [33,34]. Furthermore, characteristic peaks for S^2-^ in SnS were observed at 161.96 eV (S 2p_1/2_) and 160.78 eV (S 2p_3/2_) [33,34]. It should be noted that SnS_2_ show the energy states of Sn^4+^ with Sn 3d_3/2_ (495.12–495.3 eV) and Sn 3d_5/2_ (486.86–486.8 eV) energy levels, while S^2-^ ions show signals for S 2p_1/2_ (162.9 eV) and S 3p_3/2_ (161.7 eV) [9,23]. The presence of signals for Sn^2+^ and the absence of signals for Sn^4+^ confirm that only the Sn^2+^ state is present in SnS-80 thin films annealed below 300 °C. These results are consistent with the XRD and Raman results for SnS-80, SnS-H250, and SnS-80-H300 (Figure 4d,e), which did not show any peaks characteristic of the SnS_2_ phase.

The formation of Sn and/or S vacancies in the SnS-80, SnS-80-H250, and SnS-80-H300 thin films was investigated by calculating the Sn/S atomic ratios of these samples from the Sn 3d and S 2p peaks. The results in Table 1 show that the Sn/S ratio changed from 0.52 for SnS-80 to 2.16 for SnS-80-H300. Since the XPS examinations revealed that all the SnS-80 samples have a stoichiometry of SnS, or an Sn/S ratio of one, the ratio change with annealing should be taken qualitatively. The reduction in the amount of S in the annealed samples can be explained by the reaction between S and oxygen at temperatures above 250 °C, which leads to the loss of S from the SnS thin films [35]. A similar oxidation reaction can take place for Sn, but the temperatures may not be high enough to detect a noticeable amount. Tin-based oxides (SnO_x_) were not observed by XRD spectra, Raman spectra, and absorbance measurement (Figure 4d,e,f). Therefore, we conclude that the formation of tin oxides is insignificant and exhibits a negligible impact on the sensor properties. The reduced absorbance of the SnS-80-H300 thin film (Figure 4f) and its higher transparency (Appendix A) can be explained by the decrease in the thickness of the thin film upon heat treatment, which is due to the reactions of Sn and S with oxygen, as discussed. The presence of pinholes, as observed in Figure 6d, is further evidence for the significant losses of S and Sn due to reaction with oxygen. It is known that SnS shows p-type properties when the amount of S overwhelms that of Sn, while an excess of Sn atoms leads to a change in the conductivity of SnS from p-type to n-type [12,13,15].

In this study, the p-type and n-type conductivities of SnS thin films were determined by Hall effect measurements (Table 2). The results show that the SnS-80 and SnS-80-H250 thin films show p-type conductivities, while SnS-80-H300 shows n-type conductivity. These results are also consistent with their NO_2_-sensing behaviors, as shown later (Table 1**)**. In conclusion, while the effect of other phases was proven to be insignificant, the formation of S vacancies is the key reason for the change in the conductivity of SnS thin films from p-type to n-type, thereby strongly changing the NO_2_-sensing behavior of the annealed SnS thin films.

### 3.2. Effects of Sn and S Vacancies on Gas-Sensing Performance

A comparison of SnS-80, SnS-80-H250, and SnS-80-H300 thin-film sensors was conducted at different NO_2_ concentrations under UV illumination at RT, as shown in Figure 7a. It should be noted that the base resistance of the SnS-80 thin film was very high (~1 GΩ), indicating its low hole concentration and/or mobility (Table 2). Meanwhile, Hall effect measurements revealed an interesting observation. While SnS-80 and SnS-80-H300 exhibited clear p and n conduction types, respectively, SnS-80-H250 showed an unclear conduction type, fluctuating between n- and p-types. This may indicate a highly compensated doping state in SnS-80-H250, or the coexistence of electron and hole carriers in the material. The high resistance of SnS-80 might be derived from a relatively poor crystalline nature with inactivated dopants in the as-deposited thin film. The decreasing resistance with heat treatment (~400 MΩ for 250 °C and ~80 MΩ for 300 °C) can be explained by enhancement in the dopants activation and crystallinity. However, the conductance of SnS-80-H300 was motivated through the conduction type change from p-type to n-type due to S deficiency, as confirmed in Table 1 and Table 2.

As for sensing, the SnS-80-H300 sensor showed a measurable sensing signal toward low NO_2_ concentrations (<5 ppm) while both SnS-80 and SnS-80-H250 sensors showed no signal due to the very high resistances (Figure 7a). It is interesting, however, to observe that the as-deposited SnS sensor of the higher resistance exhibited p-type sensing at a high concentration (Figure 7d) while the SnS-80-H250 sensor of the lower resistance with compensation still presented a nil response (data not shown). The compensation of the n-type sensor signal and p-type sensor signal may lead to this result, but this needs further study.

Generally, our gas-sensing observations can be summarized as follows:(i)The change in conductivity from p-type to n-type resulted in higher sensitivity and shorter recovery time, particularly in SnS thin-film sensors annealed at 300 °C (Figure 7a,b and Appendix A).(ii)The as-deposited SnS sensors showed high base resistances, low responses, and long recovery times (Figure 7d and Appendix A).(iii)The SnS-80-H250 thin-film sensor exhibited a nil response at all the NO_2_ concentrations used.(iv)Film thickness strongly affected the response values and recovery times for both as-deposited and annealed SnS thin-film sensors (300 °C).

The response values of the annealed SnS sensors (300 °C) were larger than those of the as-deposited SnS sensors, although the NO_2_ concentrations used for the latter (25–100 ppm) were higher than those used for the former (1–5 ppm) (Figure 7c and Appendix A). Furthermore, at the optimal thickness (80 nm), the recovery time of the SnS-80-H300 sensor was approximately 12 min, while that of SnS-80 was approximately 1 h. Thus, there is a complete difference in gas-sensing properties when the conductivity of the SnS thin film changes. As discussed above, no new phases were detected when the thin films were annealed at different temperatures (250 and 300 °C). Therefore, Sn and S vacancies are responsible for these sensing differences. To further elucidate the enhanced sensing mechanism, the adsorption of NO_2_ molecules on various SnS monolayers (with and without vacancies) was simulated using DFT, as shown in Figure 8. Three monolayer structures were built for consideration as follows: a SnS monolayer with no defects (SnS); a SnS monolayer with Sn vacancy (SnS-Sn); and a SnS monolayer with S vacancy (SnS-S).

In this simulation, the adsorption energies (E_ads_) between the NO_2_ molecules and SnS monolayers were defined as:E_ads_ = E_SnS-gas_ − (E_SnS_ + E_gas_)(1)
where E_SnS-gas_ is the total energy of the SnS-NO_2_ system and E_SnS_ and E_gas_ are the energies of the SnS monolayer and NO_2_ molecule, respectively. By this definition, a smaller E_ads_ value indicates a more favorable adsorption process for the NO_2_ molecule onto the monolayer. In addition, the net electron transfer (ΔQ) between the SnS monolayer and NO_2_ molecule was calculated by:ΔQ = Q_gas–ads_ − Q_gas-iso_(2)
where Q_gas-ads_ and Q_gas-iso_ are the valence electrons of the NO_2_ molecule in the adsorbed and isolated states, respectively. By this definition, a positive value suggests an electron transfer from the SnS monolayer to the NO_2_ molecule. Table 3 summarizes the calculated energies for the adsorption of NO_2_ molecules on various SnS monolayer structures. The adsorption energies for NO_2_ molecules onto SnS, SnS-Sn, and SnS-S monolayers were approximately −0.68, −1.85, and −1.88 eV, respectively. The data reveal that NO_2_ is readily adsorbed onto SnS monolayers with vacancies, whereas it is less readily adsorbed onto the perfect SnS monolayer. These results are in good agreement with other reports on enhanced gas adsorption on layered structures with defects [9,10,36,37,38].

In addition, we calculated the charge density differences between NO_2_ molecules and various SnS monolayers. It should be noted that the electron transfer between NO_2_ and vacancy-SnS monolayers was higher than that for pristine SnS. Moreover, the highest electron transfer was obtained with the SnS-S monolayer. Generally, the SnS-S monolayer was favorable for achieving a high response value owing to the combination of low adsorption energy and large electron transfer between NO_2_ and SnS-S. These results are consistent with the observations of gas sensing shown above. In particular, the SnS-80-H300 sensor showed detecting ability at low NO_2_ concentrations (<5 ppm), unlike the other sensors. While the better sensing ability of the SnS-80-H300 sensor can be explained in terms of the more favorable adsorption energy and electron transfer, its shorter recovery time is due to the presence of pores, which enhance the gas diffusion process (Figure 6d). It should be noted that the SnS-80-H250 sensor showed a nil response to NO_2_. Therefore, the enhanced response of the SnS-80-H300 sensor should be attributed more to the effect of vacancies than to that of the pores formed at high annealing temperatures.

Apart from the decisive effect of the vacancies on the sensing behaviors of the SnS sensors as investigated above by DFT, the effect of film thickness on the sensing properties of the sensors was significant. Both the as-deposited sensors and those annealed at 300 °C showed the same trend: the response decreased when the thickness of the SnS thin-film increased. The response of SnS-30-H300 was significantly higher than that of the other sensors (Figure 7c). Similarly, the responses of the SnS-30 and SnS-50 sensors were outstanding compared to those of the SnS-80 and SnS-100 sensors (Appendix A, Supporting Information). These results can be explained by the nano-size effect. If the thickness of the sensing film approaches a value comparable to the depletion layer width, the sensor will be expected to have the highest response. The further away from optimal thickness this value is, the lower the response is. In addition, comparing the response curves of the sensors annealed at 300 °C also indicates the significant impact of particle size on the recovery times (Figure 7b). Clearly, the particle size increased gradually from SnS-30-H300 to SnS-100-300 owing to the initial difference originating from the deposition process of these thin films (Figure 2a–d). A larger particle size is always accompanied by a higher porosity, which is an advantage for the gas diffusion process. Therefore, the SnS-30-H300 sensor exhibited the slowest recovery, while the recovery of the SnS-100-H300 sensor was the fastest. In order to satisfy both the response and recovery time required for real-world applications of a gas sensors, the SnS-80-H300 sensor was chosen for further investigation.

### 3.3. Gas-Sensing Performance of a Flexible SnS Sensor and the Effect of Humidity

An interesting property of SnS thin films on PI substrates is its negligible deformation under heat treatment. Appendix A indicates that SnS-80-H300 was not completely deformed in comparison to SnS-80 under observation by the naked eye. The result reveals a similar thermal expansion coefficient between the prepared SnS thin film and the PI substrate, which is favorable for the development of a flexible sensor. Currently, one of the most important obstacles to the development of flexible sensors is their deformation at high annealing temperatures owing to the large difference in the thermal expansion coefficients of the sensing layer and the flexible substrate. Apart from low deformation at high temperature, the sensors also exhibited excellent adhesion between the SnS sensing layer and PI substrate. The gas-sensing properties of the flexible SnS-80-H300 sensor were examined at different bending angles (Figure 9a). The base resistance of the sensor increased proportionally with the bending angle (Figure 9b). However, this change in base resistance is small and can be explained by the reduction in contact between the SnS particles and/or between the SnS and Pt electrodes when the bending angle increases. The deformation does not affect the recovery process of the sensor, but the response of the sensor changes slightly. However, overall, the gas-sensing properties and the base resistance of the sensor remained similar for bending and non-bending modes. The flexible sensor also showed good short-term repeatability as well as reliable long-term stability over 60 days (Figure 9c). Furthermore, we observed that the sensor is very selective to NO_2_ gas. The sensor showed a nil response toward 5 ppm SO_2_, H_2_S, and NH_3_ gases. The results are similar to those for H_2_, CH_4_, CO, and CO_2_ gases, although the concentrations used for these gases were 20–200 times higher than that for NO_2_ (Figure 9d). These results suggest that the flexible SnS sensor fabricated in this study is stable upon bending. Therefore, it can be further improved to integrate into portable and wearable devices. For comparison, the NO_2_ sensing performances of some reported flexible sensors are summarized in Table 4 [39,40,41,42,43,44]. The SnS thin-film sensor reveals relatively short response/recovery times compared to others. However, the performance of the SnS sensor still needs to be improved to qualify for the technical demands of a real room-temperature sensor.

The effect of relative humidity (RH) on the response of the SnS-80-H300 sensor at RT was also examined, as summarized in Figure 9e,f. The base resistance of the sensor was reduced continuously owing to the electron donation by water molecules into the SnS film. This trend is similar to our previous observations [18,45]. It should be noted that the response of the SnS film exhibited a decrease onset at 30% RH. However, in the previous study, the In_2_O_3_/Rb_2_CO_3_ sensor showed a decrease onset at approximately 45% RH. These results indicate that a film with low porosity would exhibit a decrease in response quickly under exposure to RH. The presence of RH decreases the depletion layer width, which is the main reason for the significant modulation of the sensor response by the nano-size effect. Therefore, the optimal response was observed at ~30% RH. Meanwhile, the significantly lower response at high RH, especially that at 80% RH, can be explained by the formation of a water molecule layer on the SnS film surface, hindering the adsorption of NO_2_. Generally, the n-type SnS thin-film sensor exhibited good sensing performance against bending, high adhesion, and a thermal expansion coefficient that is compatible with that of a PI substrate. Therefore, our study paves the way to develop RT flexible sensors with enhanced gas-sensing performance and good bending ability.

## 4. Conclusions

In summary, we have developed a flexible sensor based on a SnS thin film and a PI substrate. The sensor shows the potential for application to selective NO_2_ sensing under bending at RT under UV illumination. Furthermore, the SnS sensing layer exhibits good adhesion ability and a thermal expansion coefficient appropriate for the PI substrate, which allows the sensor to be heated to 300 °C without deformation. This is a great advantage for improving SnS sensors on PI substrates in the next stages using techniques that require heat treatment. Furthermore, using different experimental and theoretical methods, we revealed that S vacancies play an important role in the sensing properties. Thus, our study provides a strategy for the construction of SnS-based nanosensors that can be integrated into wearable and/or flexible devices.

## Figures and Tables

**Figure 1 sensors-20-05701-f001:**
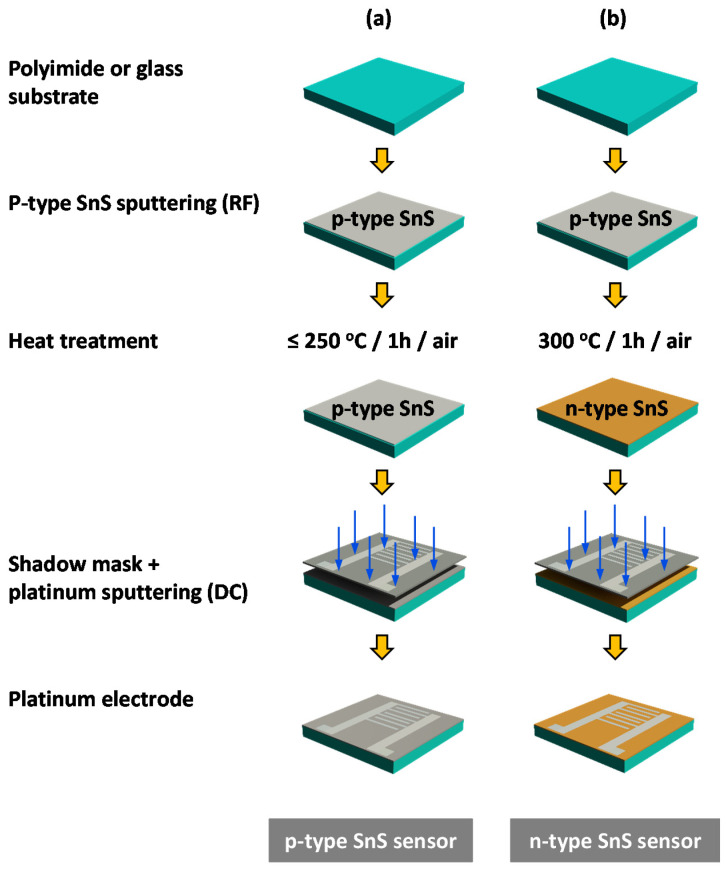
Illustration of the fabrication of tin sulfide (SnS) thin film sensors by sputtering: (**a**) a p-type SnS sensor, and (**b**) an n-type SnS sensor.

**Figure 2 sensors-20-05701-f002:**
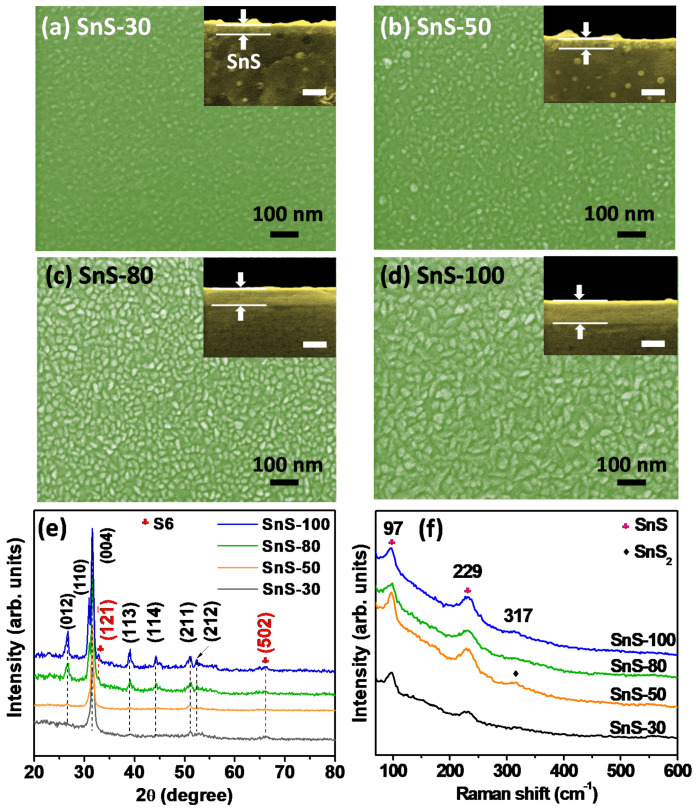
(**a**–**d**) SEM surface images of different as-deposited SnS thin films. The insets show cross-sectional SEM images with the corresponding thicknesses (the scale bar represents 100 nm). (**e**) X-ray diffraction spectra and (**f**) Raman spectra of as-deposited SnS thin films.

**Figure 3 sensors-20-05701-f003:**
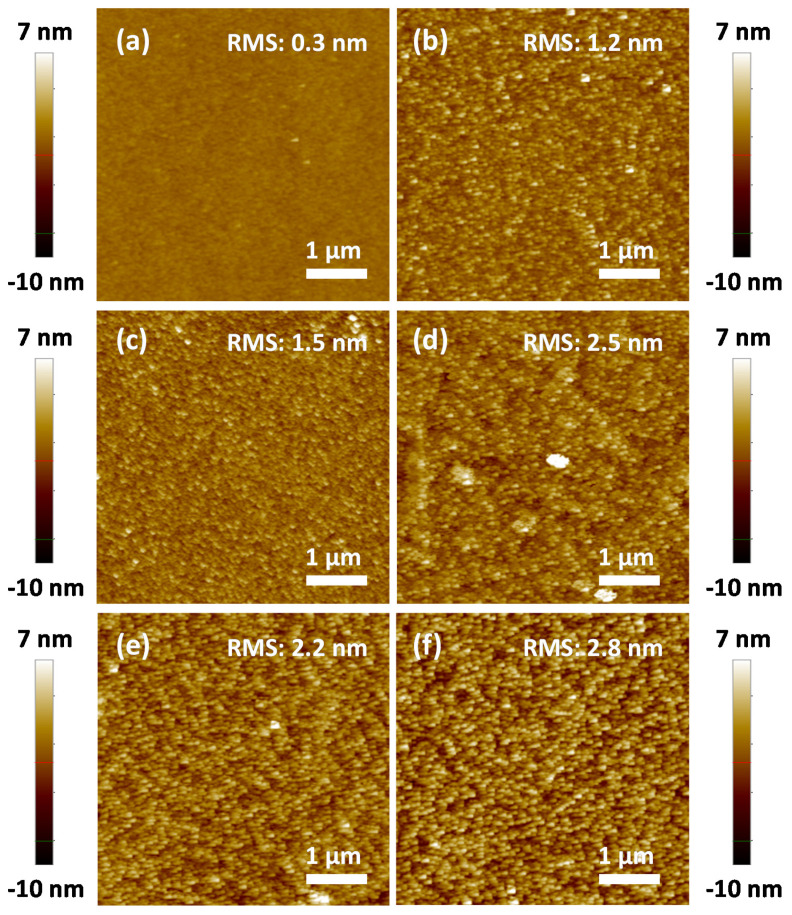
(**a**–**d**) AFM surface images of different as-deposited SnS thin films. (**a**) SnS-30, (**b**) SnS-50, (**c**) SnS-80, and (**d**) SnS-100. (**e**,**f**) AFM surface images of (**e**) SnS-80-H250 and (**f**) SnS-80-H300.

**Figure 4 sensors-20-05701-f004:**
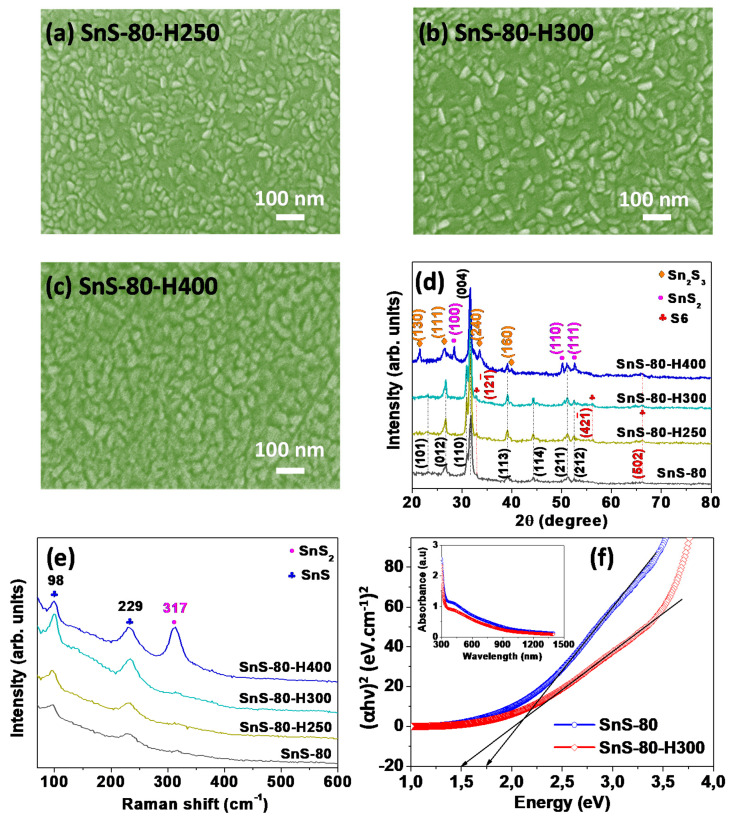
(**a**–**c**) SEM surface images of SnS-80 thin films annealed at different temperatures (250–400 °C): (**a**) 250 °C (SnS-80-H250), (**b**) 300 °C (SnS-80-H300), and (**c**) 400 °C (SnS-80-H400). (**d**) Corresponding XRD and (**e**) Raman spectra of SnS-80 thin films annealed at different temperatures (250–400 °C) and that of the as-deposited SnS-80 thin film. (**f**) Tauc plots for the as-deposited SnS-80 and SnS-80-H300 thin films. The inset shows the corresponding absorbance curves.

**Figure 5 sensors-20-05701-f005:**
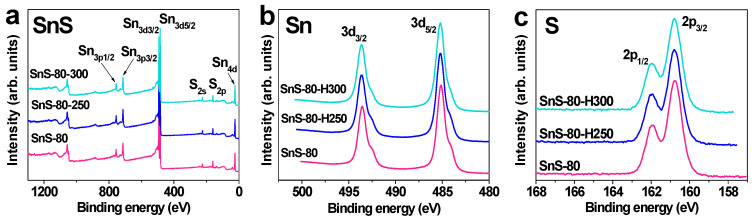
Survey (**a**) and high-resolution ((**b**) Sn 3d and (**c**) S 2p) XPS profiles of SnS-80, SnS-80-H250, and SnS-80-H300.

**Figure 6 sensors-20-05701-f006:**
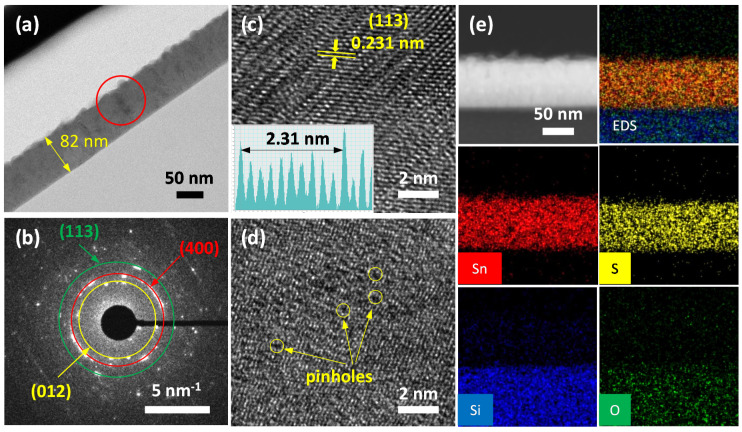
(**a**) TEM image of a SnS-80-H300 thin film on a glass substrate. (**b**) SAED trace of the selected region in (a) demonstrating the polycrystalline structure of the thin film. High-resolution TEM images of different regions (**c**) without and (**d**) with pinholes. The inset in (c) shows the distance between (113) lattice plane fringes. (**e**) Elemental mapping results for a SnS-80-H300 thin film on a glass substrate.

**Figure 7 sensors-20-05701-f007:**
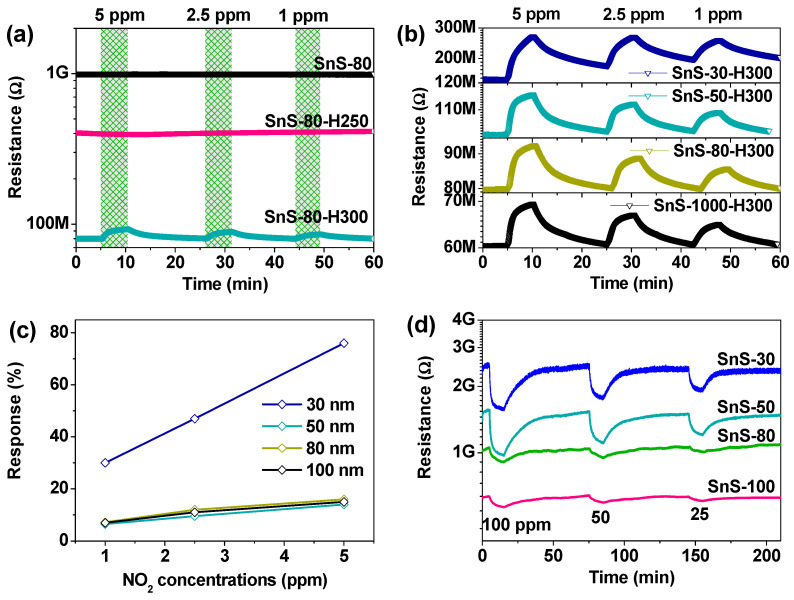
(**a**) Real-time resistance curves obtained under different NO_2_ concentrations for as-deposited SnS-80, SnS-80-H250, and SnS-80-H300 sensors. (**b**) Response curves for various concentrations of NO_2_ gas of SnS sensors with different thicknesses annealed at 300 °C. (**c**) Corresponding responses of the sensors as derived from (b). (**d**) Dynamic response curves for as-deposited SnS thin-film sensors with different thicknesses showing p-type sensing behavior toward high-concentration NO_2_ gas. All gas-sensing measurements were conducted under UV illumination at RT (25 °C).

**Figure 8 sensors-20-05701-f008:**
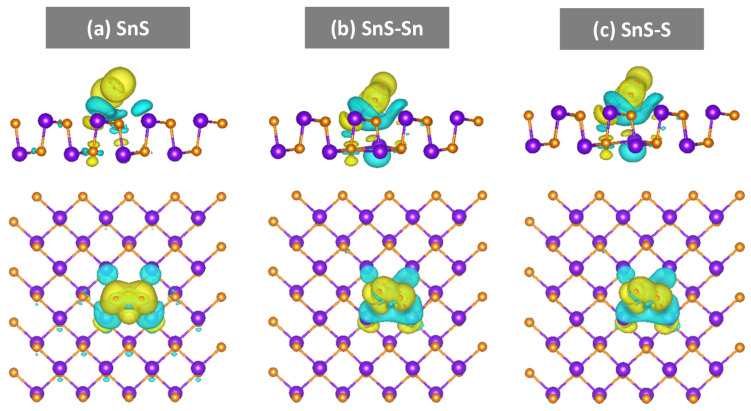
Side-view (top) and top-view (bottom) images showing the charge density difference of NO_2_ molecules adsorbed on various SnS monolayer structures: (**a**) SnS, (**b**) SnS-Sn, and (**c**) SnS-S. The yellow and cyan regions represent the areas of electron accumulation and depletion, respectively.

**Figure 9 sensors-20-05701-f009:**
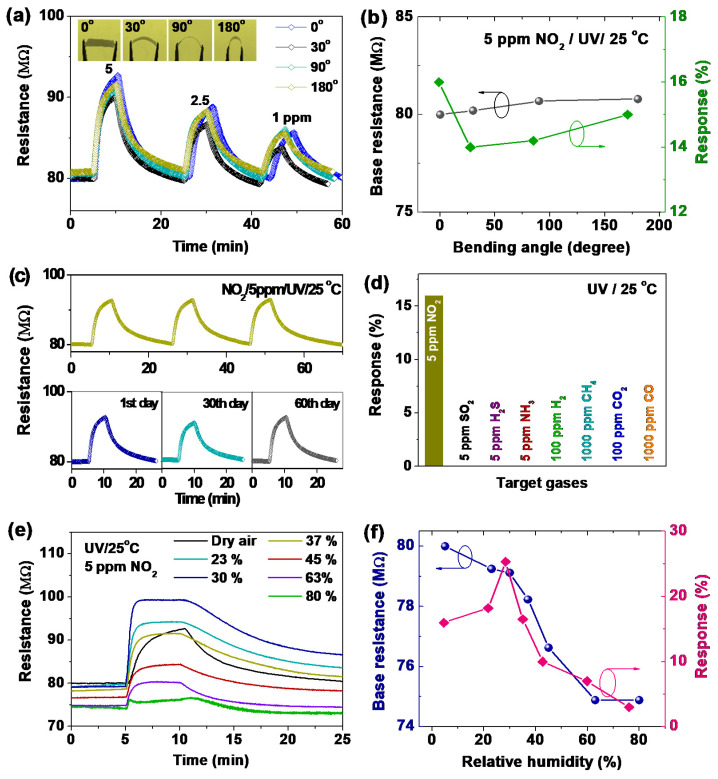
(**a**) Real-time resistance curves of SnS-80-H300 recorded under different NO_2_ concentrations and with different bending angles. (**b**) Effect of bending angle on the base resistance and response of the sensors derived from (a). (**c**) Short-term stability of the SnS-80-H300 sensor over three cycles (top) and corresponding long-term stability (bottom) of the SnS-80-H300 toward 5 ppm NO_2_ measured over 60 days. (**d**) Relative responses of SnS-80-H300 to various target gases showing the high selectivity of the sensor toward NO_2_. (**e**) Real-time response of the SnS-80-H300 sensor to 5 ppm NO_2_ under different RH conditions. (**f**) Effect of RH on the base resistance and response of the sensor. All gas-sensing measurements were conducted under UV illumination at RT (25 °C).

**Table 1 sensors-20-05701-t001:** Summaries of the atomic percentages of Sn and S and Sn/S atomic ratios in SnS-80 thin films annealed at different temperatures.

Samples	Sn (at%)	S (at%)	O (at%)	C (at%)	Sn/S
As-deposited SnS-80	12.29	23.82	21.70	42.19	0.52
SnS-80-H250	12.28	11.47	32.39	43.86	1.07
SnS-80-H300	18.01	8.33	49.75	23.91	2.16

**Table 2 sensors-20-05701-t002:** Summaries of the electrical properties of various SnS thin films as obtained by Hall effect measurement.

Samples	Conductivity (Ω^−1^cm^−1^)	Carrier Concentration (cm^−3^)	Mobility (cm^2^/V.s)	Carrier Type
As-deposited SnS-80	3.65 × 10^−5^	3.83 × 10^12^	59.5	p-type
SnS-80-H250	-	-	-	-
SnS-80-H300	4.3 × 10^−4^	3.7 × 10^13^	72.5	n-type

**Table 3 sensors-20-05701-t003:** Calculated adsorption parameters for NO_2_ on various SnS monolayer structures.

Monolayer Structures	E_ads_ (eV)	ΔQ (e)
SnS	−0.68	1.03
SnS-Sn	−1.85	1.34
SnS-S	−1.88	1.46

**Table 4 sensors-20-05701-t004:** Performances of several flexible sensors toward NO_2_ gas.

Materials	[NO_2_]/ppm	T/°C	Response (%)	t_res_/t_rec_	Reference
^a^ CSA-PPy	100	RT	36	250 s/40 min	[39]
^a^ PPy-ZnO	10	RT	15	10 min/3 h	[40]
^a^ rGO/CeO_2_	10	RT	20	10 min/10 min	[41]
^b^ CNT/reduced graphene	5	RT	13	-/over 60 min	[42]
^b^ SnS_2_ flakes/PI	1	RT	50	8 min/30 min	[43]
^b^ SnS_2_ flakes/Al_2_O_3_/PI	5	RT	309	8 min/over 30 min	[43]
^b^ Graphene	100	RT	26	5 min/over 15 min	[44]
SnS thin-film	5	RT	18	5 min/12 min	This study

a: Response = (R_a_-R_g_)/R_a_ * 100 %; b: Response = (R_g_-R_a_)/R_a_ * 100 %; RT: room temperature.PPy: Polypyrrole; PI: Polyimide; CSA: Camphor sulfonic acid.

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
