# Peer review of "Defect-Induced Gas-Sensing Properties of a Flexible SnS Sensor under UV Illumination at Room Temperature"

_sensors, 2020, doi:10.3390/s20195701_

Round 1
Reviewer 1 Report
The authors, Nguyen Manh Hung et al. have studied the effects of annealing on the morphology and gas-sensing properties of SnS thin films, fabricated by RF sputtering, and have demonstrated that S vacancies produce n-type semiconductor behavior in the films and therefore better sensor performance under UV illumination at RT for NO2 sensing. The sensor is very selective to NO2 compared to other gases. Additionally, DFT calculations were used to simulated NO2 adsorption for different SnS configurations (No defects, Sn vacancies, and S vacancies). Finally, the authors have shown to successfully develop a flexible sensor by depositing SnS thin film on a polyimide (PI) substrate.
Broad Comments:
- There is a feature on each spectrum of Sn, at low binding energies, that is not discussed in the manuscript. It seems like the feature is decreasing with annealing. Could the authors explain this feature? Features usually mean more than one type of the analyzed element is present, and since XPS is a surface-sensitive technique, this type of Sn could be on the surface and not be detected by Raman due to its detection limit. Although the feature is a lower binding energy compared to metal Sn, I am wondering if this is due to the calibration performed to the data, which could lead to a bad interpretation. Could the authors elaborate more on how the calibration was done, and how much the peaks were shifted?
- The concentration of O is for the SnS-80-H300 sample is high. It is known that annealing in air could increase the oxidation of the samples, but this is not discussed in the manuscript. And, although the presence of S vacancies is a well-accepted explanation for the n-type behavior, it is also known that SnOx will act as a self-passivation layer and therefore it could affect the performance of the sensor. Therefore, more discussion is needed respect to the possible presence of SnOx.
Specific Comments:
- Line 224: it should be “Fig 4 (d, e)” instead of “Fig 4 (e, f)”. Figure f shows Tauc Plots.
- Line 232: authors mention that all the SnS-80 samples have a stoichiometry of SnS, but actually, XPS data shows that it is a stoichiometry SnS2. Could the authors elaborate more on this?
- It could be useful to have the information about atomic concentrations for all the samples annealed at 300 C, this could be provided in the SI document. Specially oxygen content.
Author Response
Reviewer # 1:
Thanks for your questions and comments.
Question 1.
There is a feature on each spectrum of Sn, at low binding energies, that is not discussed in the manuscript. It seems like the feature is decreasing with annealing. Could the authors explain this feature? Features usually mean more than one type of the analyzed element is present, and since XPS is a surface-sensitive technique, this type of Sn could be on the surface and not be detected by Raman due to its detection limit. Although the feature is a lower binding energy compared to metal Sn, I am wondering if this is due to the calibration performed to the data, which could lead to a bad interpretation. Could the authors elaborate more on how the calibration was done, and how much the peaks were shifted?
Answer:
Yes. We completely agree with your assessment of the presence of a small feature on each spectrum of Sn. Considering the XPS data of Sn and its compounds in the literature [1], Sn metal shows the binding energy of Sn 3d5/2 in a range from approximately 484.6 eV to 485.15 eV. This value strongly increases when Sn exists in compounds such as SnS, SnS2, SnO, and SnO2. Therefore, in this study, we believe that the small features that you mentioned display the presence of a small amount of Sn metal, which could be originated from target fabrication. The decrease of these features should be understood by the reaction of Sn with S and/or O at higher temperatures [2]. Therefore, the feature decreases with annealing. However, the presence of Sn metal is insignificant as proved by other analysis techniques as well as the high-resistance of the as-synthesized SnS thin-films.
As we indicated in the experimental section, XPS peaks were calibrated using the binding energy of C 1s (284.8 eV). Accordingly, each high-resolution spectrum of Sn was accompanied by a C 1s peak measured in the same condition with the Sn high-resolution spectrum. The C 1s peaks are shown in the following figure:
These C 1s values are used to compare to the value of 284.8 eV, which determined shifts are 0.1, 0.3, and 0.77 eV to SnS-80, SnS-80-H250, and SnS-80-H300, respectively. These shift values are used to calibrate the corresponding spectra of Sn.
Question 2.
The concentration of O is for the SnS-80-H300 sample is high. It is known that annealing in air could increase the oxidation of the samples, but this is not discussed in the manuscript. And, although the presence of S vacancies is a well-accepted explanation for the n-type behavior, it is also known that SnOx will act as a self-passivation layer and therefore it could affect the performance of the sensor. Therefore, more discussion is needed respect to the possible presence of SnOx.
Answer:
Thanks for your comment. The results in Table 1 (page 10) showed that the high O concentration in SnS-80-H300 comes with a strong decrease of C concentration. However, we agree that the reaction between tin and oxygen can take place, but the oxide phase was not detected by the measurements. We discussed this issue in lines 240-244 with highlighted text (page 9).
Specific Comments
- Line 224: it should be “Fig 4 (d, e)” instead of “Fig 4 (e, f)”. Figure f shows Tauc Plots.
Answer:
Thanks for indicating the mistake. We corrected the mistake in the revised manuscript with the highlight (line 229).
- Line 232: authors mention that all the SnS-80 samples have a stoichiometry of SnS, but actually, XPS data shows that it is a stoichiometry SnS2. Could the authors elaborate more on this?
Answer:
Thank you for your comment. As we indicated in Fig. 2 (e), apart from the SnS phase, amount of hexa-sulfur (S6) is observed in as-synthesized samples. At 250 oC, S6 phase is significantly burned, which is proved by a sudden decrease of S contents in SnS-80-H250 compared to SnS-80 (table 1). In the meantime, XRD and Raman results confirmed that there are no SnS2 phase in as-synthesized SnS-80. Therefore, we believe that XPS data showed a stoichiometry SnS2 should be considered by the attribution of the S6 phase.
- It could be useful to have the information about atomic concentrations for all the samples annealed at 300 C, this could be provided in the SI document. Specially oxygen content.
Answer:
Thanks for your suggestion. We understand that the information about atomic concentrations for all the samples annealed at 300 oC is very useful for a statistic and comprehensive analysis if they are available. However, we think that the XPS technique is sensitive to a depth of approximately 10 nm from the surface. In the meantime, all samples were synthesized and annealed in the same conditions. Therefore, samples with different thicknesses are expected to show similar atomic concentrations.

Reviewer 2 Report
The authors fabricated a flexible NO2 gas sensor that was made of SnS film.
The following questions should be revised for improving the quality of the paper.
- In P. 11, lines from 264 to 279, the typing is italic type.
- In the typing, the authors should check the single space or double space through text.
- How about the long-term stability of the flexible NO2 gas sensor that was made of SnS film.
- Many papers reported the flexible NO2 gas sensor so that the authors should compare the performance of the present work NO2 gas sensor with the NO2 sensors in the literature.
- The authors fabricated a flexible NO2 gas sensor that was made of the SnS film. They clearly studied the NO2-sensing mechanism under UV irradiation using DFT and effect of the thickness and sintering temperature on the response of the flexible sensor. However, some typos were found in the manuscript. Additionally, the authors should clearly explain the reason for choosing SnS as the sensing film compared with others metal oxides such as SnO2, WO3.
Author Response
Thanks for your questions and comments.
Question 1.
In P. 11, lines from 264 to 279, the typing is italic type.
Answer:
Thanks for indicating the mistake. We corrected the mistake with the highlight. They are now from lines 272 to 291 in the revised manuscripts.
Question 2.
In the typing, the authors should check the single space or double space through text.
Answer:
Yes. We modified the whole manuscript per your suggestion.
Question 3.
How about the long-term stability of the flexible NO2 gas sensor that was made of SnS film.
Answer:
We checked the long-term stability of the SnS thin-film sensor after one and two months, and the results were shown in Fig. 9 (c).
Question 4.
Many papers reported the flexible NO2 gas sensor so that the authors should compare the performance of the present work NO2 gas sensor with the NO2 sensors in the literature.
Answer:
Thanks for your valuable recommendation. A statistic table and related discussions have been added to the revised manuscript (page 15).
Question 5.
The authors fabricated a flexible NO2 gas sensor that was made of the SnS film. They clearly studied the NO2-sensing mechanism under UV irradiation using DFT and effect of the thickness and sintering temperature on the response of the flexible sensor. However, some typos were found in the manuscript. Additionally, the authors should clearly explain the reason for choosing SnS as the sensing film compared with others metal oxides such as SnO2, WO3.
Answer:
Thanks for indicating the mistake. We check through the manuscript and some typos have been corrected with highlights.
Yes. We totally agree with the reviewer that many thin-films of other metal oxides can be used as sensing-layers on various flexible substrates. However, the development of these flexible sensors is hindered by a heat-treatment process which is always used to enhanced the stability of the metal oxide sensors. However, metal oxides-based flexible sensor is strongly deformed after heat treatment owing to a big difference in the coefficient of thermal expansion between the MOS-sensing layers and the flexible substrates. In this study, apart from short response/recovery times, SnS thin-film exhibited no deformation by heat up to 300 oC. Furthermore, the sensing layer shows good adhesion ability under various bending modes. These properties are outstanding compared to that of metal oxides-based flexible sensors. We have added these discussions to the introduction part with the highlighted text.
Reviewer 3 Report
Your sensor is actually robust sensor that reacts selectively to NO2, and I think this research value is high. But I have some suggestions for you to revise your paper.
(1) Line 151-152: I recommend you should check again the scale bar of AFM and SEM result to show the good agreements.
(2) Line 183-184: The crystallite size and domain size can be estimated by using XRD. You can utilize some equations from this: http://prism.mit.edu/XRAY/oldsite/CrystalSizeAnalysis.pdf
(3) Figure 2f and 4e: I recommend that you apply the fitting of the result value using the Origin S/W to the graphics for a clearer Raman peak comparison. It would also be better if an analysis of the degree of intensity of the Raman spectrum is added.
(4) Figure 7 and 9: Speaking of response time and recovery time, RGO sensor has 10 seconds, Zinc stannate film sensor has 8 and 52 seconds, polyaniline/CNT composite sensor has a 1.8 and 5.2 seconds. For commercialization, I recommend you consider ways to reduce the recovery time (over 10 mins) and response time (~5 mins) of your sensor. Likewise, most humidity sensors have a response time between 30 and 60 seconds, or a very short response time. Your sensor has a very long response time and recovery time compared with them, Do you have any way to improve this problem?
Author Response
Thanks for your question and your comment.
Question 1
Line 151-152: I recommend you should check again the scale bar of AFM and SEM result to show the good agreements.
Answer:
Thanks for your valuable suggestion. AFM images have been replaced by the better ones. The corresponding scale bars have been corrected as well (Fig. 3).
Question 2
Line 183-184: The crystallite size and domain size can be estimated by using XRD. You can utilize some equations from this: http://prism.mit.edu/XRAY/oldsite/CrystalSizeAnalysis.pdf.
Answer:
Thanks for your comment and your valuable document. We calculated crystalline size for various samples including SnS-80, SnS-80-250, and SnS-80-H300. The detail results are shown in Fig. S3, and we find out that the crystalline sizes calculated are smaller than those observed by SEM and AFM. This result suggests that the particles are polycrystals rather than single crystals. The result also reveals that size-growths at high temperatures in SnS thin-films. Some discussions were added to the manuscript (lines 189 – 192, page 7).
Question 3
Figure 2f and 4e: I recommend that you apply the fitting of the result value using the Origin S/W to the graphics for a clearer Raman peak comparison. It would also be better if an analysis of the degree of intensity of the Raman spectrum is added.
Answer:
Thanks for your recommendation. We agree with that removing the background noise can resolve the peaks more clearly. However, this manipulation may raise ambiguity for very small intensity peaks rather than making clear the peaks. Since our discussion of Raman results are limited to supporting the identification of the phases, we would rather provide the as-measured plots, which are acceptable for the judgement.
Question 4
Figure 7 and 9: Speaking of response time and recovery time, RGO sensor has 10 seconds, Zinc stannate film sensor has 8 and 52 seconds, polyaniline/CNT composite sensor has a 1.8 and 5.2 seconds. For commercialization, I recommend you consider ways to reduce the recovery time (over 10 mins) and response time (~5 mins) of your sensor. Likewise, most humidity sensors have a response time between 30 and 60 seconds, or a very short response time. Your sensor has a very long response time and recovery time compared with them, Do you have any way to improve this problem?
Answer
Yes. We know that the response/recovery times of the current sensor are relatively longer and need to be improved. This problem is also one of main obstacles to develop RT gas sensors in general. We found the light energy at RT is not sufficient to measure a seconds-range response kinetics. We have to put this issue as a future research.
Reviewer 4 Report
Comments
The authors reported annealing process to enhance the gas sensing capability of SnS thin-film based gas sensors for portable and flexible SnS sensor. SnS thin-film sensors usually do not show ideal performance duo to high resistance, low response, and long recovery time of the SnS films.
The electronic and gas-sensing properties of SnS strongly depend on its surface defects. The authors annealed SnS thin films of different thickness at varied annealing temperatures. They demonstrated that SnS thin films showed p-type semiconducting nature in general, but turned into n-type behavior after being annealed at 300 °C. They studied the films before and after the annealing processes by both surface analysis experiments and computational simulation, and came to the conclusion that the change was attributed to the appearance of S vacancies on the film surface. The presence of S vacancies made the n-type SnS sensor with better sensing performance under UV illumination at room temperature than that of a p-type SnS sensor. Good selectivity for different gasses was demonstrated in this kind of simple sensors. They also deposited n-type SnS on flexible substrate and showed some potential in real applications.
The results may have some technical value for practical applications. The manuscript could be accepted for publication after minor revision.
1, Figure 3, the AFM images should be replaced with images of better quality (less noise).
2, Since the sample preparing processes are very simple (Fig. 1), it would be much more convincing that additional samples are prepared at a fixed thickness (e.g., 80 nm) but annealed at varied temperature between 250 °C and 300 °C, with a step of 10-20 °C and more than 3 samples at one annealing temperature, to show a statistic analysis for the transition from p-type to n-type by Hall measurements. Such a statistic result will be more valuable and helpful for researchers in related community.
Author Response
Thanks for your questions and comments.
Question 1
Figure 3, the AFM images should be replaced with images of better quality (less noise)
Answer:
Thanks for your valuable suggestion. We have replaced the images by better ones.
Question 2
Since the sample preparing processes are very simple (Fig. 1), it would be much more convincing that additional samples are prepared at a fixed thickness (e.g., 80 nm) but annealed at varied temperature between 250 °C and 300 °C, with a step of 10-20 °C and more than 3 samples at one annealing temperature, to show a statistic analysis for the transition from p-type to n-type by Hall measurements. Such a statistic result will be more valuable and helpful for researchers in related community.
Answer:
Thanks for your comment. We chose a step of 50 oC for our investigation owing to the limited accuracy of the furnace we used. We think that such a statistic and comprehensive investigation is a large amount of work and it should be considered with a separate study. Furthermore, a precise temperature control of the furnace is essential to conduct this investigation.
Round 2
Reviewer 1 Report
Authors have answered all questions and have done the relevant changes on the manuscript